# Influence of Age and Gender on Nasal Airway Patency as Measured by Active Anterior Rhinomanometry and Acoustic Rhinometry

**DOI:** 10.3390/diagnostics13071235

**Published:** 2023-03-24

**Authors:** Jing-Jie Wang, Yi-Fang Chiang, Rong-San Jiang

**Affiliations:** 1Department of Otolaryngology, Taichung Veterans General Hospital, Taichung 40705, Taiwan; 2School of Medicine, College of Medicine, National Yang Ming Chiao Tung University, Taipei 112304, Taiwan; 3Department of Medical Research, Taichung Veterans General Hospital, Taichung 40705, Taiwan; 4School of Medicine, Chung Shan Medical University, Taichung 40201, Taiwan; 5Rong Hsing Research Centre for Translational Medicine, National Chung Hsing University, Taichung 40201, Taiwan

**Keywords:** acoustic rhinometry, active anterior rhinomanometry, age, gender, nasal airway patency

## Abstract

In this study, we aimed to investigate the influences of age and gender on nasal airway patency, as measured by both active anterior rhinomanometry (AAR) and acoustic rhinometry (AR). The nasal airway patency of healthy subjects was evaluated using AAR and AR. In AAR, the subjects generated airflow actively through inspiration and expiration in repetitions of 10, while nasal patency was measured at an inspiratory and expiratory reference pressure of 75 Pa. In AR, we assessed the geometry of the nasal cavity through the analysis of sound waves reflected from the nasal cavities in order to measure both cross-sectional areas and nasal volumes. The subjects were divided by gender, with all males and females then grouped by ages of 20–39 years, 40–59 years and ≥60 years. There were 40 subjects in each group. The mean resistance measured by AAR and the cross-sectional areas and nasal volumes measured by AR were not different between the different age groups; however, the cross-sectional areas and nasal volumes were found to be lesser in females than in males. Our results showed that nasal airway patency was not affected by age, while females were shown to have wider nasal passages than males.

## 1. Introduction

The nose plays an important role in combating inhaled foreign particles and detecting odorants for olfaction, while also taking in a large volume of air through the nostrils and nasal cavities [1]. However, the nasal airway itself is highly complex, having an intricate three-dimensional anatomy.

Nasal obstruction is among the most common presenting symptoms when visiting primary care clinics and may affect up to one-third of the population [2]. Although some clinicians consider that nasal obstruction implies a blockage occurring within the nasal cavity due to anatomic, physiologic and/or pathophysiologic factors, nasal obstruction is most commonly defined as a patient symptom manifested as a sensation of experiencing insufficient airflow through the nose [3]. The patency of nasal passages, mucociliary function, airflow receptors, autonomic function and degree of mucosal inflammation determines one’s optimal nasal airflow [2]. The most common clinical manifestations of nasal obstruction are the subjective sensation of congestion, stuffiness, fullness or blockage within the nose [4].

We are able to evaluate the subjective sensation of congestion, stuffiness, fullness or blockage within the nose through certain subjective evaluation tools, including the Nasal Obstruction Symptom Evaluation scale [5,6] and the 22-item Sinonasal Outcome Test [7,8]. However, objective evaluation tools have also been used to evaluate nasal function since anatomic, physiologic and pathophysiologic factors all influence nasal airway patency. Objective measurement techniques, such as rhinomanometry and rhinometry, could provide a more reliable assessment of nasal patency than the use of subjective evaluation [9].

The use of rhinomanometry became popular in 1980,when it was implemented tosimultaneously measure nasal airflow and the pressure gradient from which nasal airway resistance is calculated [10]. Rhinomanometry can be either anterior or posterior and either active or passive. Anterior active rhinomanometry (AAR) is the most commonly used rhinomanometry method as it can be more easily performed [11]. When performing AAR, testees will generate airflow actively through both inspiration and expiration in repetitions of 10 at inspiratory and expiratory reference pressures of 75 or 150 Pa.

Acoustic rhinometry (AR) is a static test and was used by Hiberg et al. in 1989 for nasal patency evaluation [12]. The geometry of the nasal cavity is assessed using sound waves, which are reflected from the nasal cavities to measure cross-sectional areas and volumes within the nasal cavities. The subject is not required to actively breathe through the nostrils.

Both AAR and AR allow doctors to evaluate nasal airway patency by measuring nasal airway resistance, cross-sectional areas of nasal cavities and nasal volumes. Nevertheless, many factors, such as anatomic or physiological change, may affect the measurements of AAR and AR. In pregnant women, the cross-sectional area of the nasal cavities decreases significantly between the first and third trimester, although there is no difference seen between each trimester with regard to nasal airway resistance [13]. The effect of age and gender on nasal airway patency has rarely been investigated. Lindemann et al. reported that both the cross-sectional areas of the nasal cavities and nasal volumes were significantly higher in the elderly than in young subjects, but no difference was seen in nasal airway resistance [14]. In this study, we aimed to investigate the influence of age and gender on nasal airway patency using both AAR and AR.

## 2. Materials and Methods

### 2.1. Study Design

This single-center prospective study was conducted at the Department of Otolaryngology, Taichung Veterans General Hospital, Taichung, Taiwan.

### 2.2. Study Subjects 

Healthy Taiwanese volunteers were recruited by pasting posters in the hospital, and they were asked about their nasal symptoms. Anyone who had the symptoms of nasal obstruction, rhinorrhea, sneezing or itchy nose was excluded from the study. If there was a history of nasal surgery, any use of drugs which influenced nasal function, such as antihistamines, steroid use within the last month or if the subject had acquired an acute nasal infection within the past week, they too were also excluded.

All eligible subjects were grouped by age and gender. The age groups for all male and female subjects were: 20–39 years, 40–59 years and ≥60 years, with 40 subjects included in each group (Figure 1). All those involved underwent both AAR and AR to measure their nasal airway patency. This study was approved by the Institutional Review Board (I) of Taichung Veterans General Hospital (protocol code CF18048A). Written informed consent was collected from all enrolled subjects.

### 2.3. Nasal Airway Patency Tests

In this study, nasal airway patency was evaluated objectively through AAR and AR, with a 10min break being given between these 2 tests.

#### 2.3.1. Active Anterior Rhinomanometry

Active anterior rhinomanometry was performed according to the guidelines of the International Committee on Standardization of Rhinomanometry using an NR6 Rhinomanometer (GM Instruments, Ltd., Kilwinning, UK) [15]. Each testee remained seated for 30 min prior to testing to better adapt to the environment. Afterwards, a face mask was worn tightly while the subject quietly continued breathing with a closed mouth in anupright sitting position. For each nostril, inspiratory nasal resistance was calculated over four inspiratory–expiratory cycles at a fixed pressure of 75 Pascal. The right, left and total nasal resistance in Pa/cm^3^/s and nasal airflow in cm^3^/s were all recorded.

#### 2.3.2. Acoustic Rhinometry

An A1 Acoustic Rhinometer (GM Instruments, Ltd., Kilwinning, UK) was used to measure the geometry of the nasal cavity [16,17]. The testee remained seated for at least 20 min prior to testing in order to acclimatize to the environment. A nose piece was then positioned parallel to the sagittal plane of the subject’s head at a 45-degree angle to the coronal plane to produce an acoustic seal without distorting the outer nose. The testee was asked to hold their breath and avoid swallowing during the acquisition of the acoustic data. Three consecutive readings were taken in order to calculate an average value, with an acoustic rhinometry curve then being generated for each nasal cavity. The following values were recorded: (1) the first minimal cross-sectional area (MCA_1_, cm^2^), (2) the second minimal cross-sectional area (MCA_2_, cm^2^), (3) the volume between the tip of the nosepiece and 3.0 cm into the nasal cavity (NV 0–3, cm^3^) and (4) the volume of the nasal cavity between 2.0 and 5.0 cm from the tip of the nosepiece (NV 2–5, cm^3^).

### 2.4. Statistical Analyses

Descriptive data are presented as mean ±standard deviation. Ages were compared between the male and female subjects using the Mann–Whitney U test. The AAR and AR values in each age group were compared between the males and females using the Mann–Whitney U test. The AAR and AR values were also compared among the 3 age groups using the Kruskal–Wallis test. The associations between inspiratory resistance, inspiratory nasal flow, MCA1, MCA2, NV 0–3, NV 2–5, age and gender were quantified using linear regression. The AAR and AR values at the 10th/90thpercentile were defined as the normative data, with those values being correlated using Spearman’s rho. All computations were performed using SPSS version 17.0 (SPSS, Inc., Chicago, IL, USA). Two-tailed *p*-values < 0.05 were considered to be statistically significant. 

## 3. Results

### 3.1. Study Subjects

There were 40 male and 40 female subjects in each of the three age groups (Figure 1). The male mean age was 26.7 ± 4.21 for the 21- to39-year-old group, 49.0 ± 6.52 for the 40- to 59-year-old group and 67.9 ± 6.85 for the ≥60-year-old group. For females, the mean age was 28.2 ± 4.86for the 21- to 39-year-old group, 50.1 ± 6.35 for the 40- to 59-year-old group and 66.4 ± 5.23 for the ≥60-year-old group. There were no differences in age between male and female subjects (*p* = 0.086, 0.375, 0.369, respectively).

### 3.2. Active Anterior Rhinomanometry

#### 3.2.1. Inspiratory Resistance

The inspiratory resistance for ARR is shown in Table 1. The difference in total inspiratory resistance was not significant between the three age groups for both male and female subjects; however, the difference in total inspiratory resistance was significant between the male and female subjects for the age group of 40–59 years (*p* = 0.001) (Figure 2).

The 90th percentile male total inspiratory resistance was 0.32 for the three age groups, 0.35 in the age group of 20–39 years, 0.26 in the age group of 40–59 years and 0.30 in the age group of ≥60 years. The 90th percentile female total inspiratory resistance was 0.37 for the three age groups, 0.27 in the age group of 20–39 years, 0.39 in the age group of 40–59 years and 0.44 in the age group of ≥60 years. 

#### 3.2.2. Inspiratory Flow

The inspiratory flow for ARR is shown in Table 2. The difference in total inspiratory resistance was not significant between the three age groups for both male and female subjects; however, the difference in total inspiratory resistance was significant between the male and female subjects for the age group of 40–59 years (*p* = 0.001) (Figure 2). Adjusted for age, the total nasal flow was significantly higher in males than females (*p* < 0.05).

The 10th percentile male total inspiratory flow was 235.72 for the three groups, 215.77 in the age group of 20–39 years, 295.41 in the age group of 40–59 years and 253.69 in the age group of ≥60 years. The 10th percentile female total inspiratory flow was 202.79 for the three age groups, 281.99 in the age group of 20–39 years, 196.72 in the age group of 40–59 years and 174.77 in the age group of ≥60 years.

### 3.3. Acoustic Rhinometry

#### 3.3.1. Cross-Sectional Area

The MCA_1_ for AR is shown in Table 3. The difference in male averageMCA_1_ was seen to be significant between the age groups of 20–39 years and 40–59years (*p* = 0.012) (Table 3). The difference in female averageMCA_1_ was seen to be significant between the age groups of 20–39 years and ≥60 years (*p* = 0.026). The male average MCA_1_ was significantly larger than the female average MCA_1_ in all three age groups (Figure 3). Adjusted for age, the average MCA_1_ was significantly higher in males than females (*p* < 0.05).

The MCA_2_ for AR is shown in Table 4. The difference in averageMCA_2_ was not significant between the three age groups for both male and female subjects, nor between the male and female subjects for all three age groups (Figure 3).

The 10th percentile male average MCA_1_ was 0.52 in the age group of 20–39 years, 0.60 in the age group of 40–59 years and 0.62 in the age group of ≥60 years. The 10th percentile female average MCA_1_ was 0.41 in the age group of 20–39 years, 0.55 in the age group of 40–59 years and 0.54 in the age group of ≥60 years. The 10th percentile male MCA_2_ was 0.20 for the three age groups, 0.26 in the age group of 20–39 years, 0.19 in the age group of 40–59 years and 0.20 in the age group of ≥60 years. The 10th percentile female MCA_2_ was 0.21 for the three age groups, 0.21 in the age group of 20–39 years, 0.25 in the age group of 40–59 years and 0.19 in the age group of ≥60 years.

#### 3.3.2. Nasal Volume

The NV 0–3 for AR is shown in Table 5. The difference in average NV 0–3 was not significant between the three age groups for both male and female subjects; however, the male average NV 0–3 was significantly larger than the female average NV 0–3 in all three age groups (Figure 4). Adjusted for age, the average NV 0–3 was significantly higher in males than females (*p* < 0.05).

The NV 2–5 for AR is shown in Table 6. The difference in average NV 2–5 was not significant between the three age groups for both male and female subjects, nor between male and female subjects in all three age groups (Figure 4). The average NV 2–5 was significantly increased with age (*p* for trend < 0.05) after being adjusted for gender.

The 10th percentile male average NV 0–3 was 1.77 for the three age groups, 1.82 in the age group of 20–39 years, 1.83 in the age group of 40–59 years and 1.56 in the age group of ≥60 years. The 10th percentile female average NV 0–3 was 1.50 for the three age groups,1.35 in the age group of 20–39 years, 1.51 in the age group of 40–59 years and 1.58 in the age group of ≥60 years. The 10th percentile male average NV 2–5 was 2.19 for the three age groups, 2.26 in the age group of 20–39 years, 1.96 in the age group of 40–59 years and 2.32 in the age group of ≥60 years. The 10th percentile female average NV 2–5 was 2.19 for the three age groups, 1.85 in the age group of 20–39 years, 2.21 in the age group of 40–59 years and 2.52 in the age group of ≥60 years.

### 3.4. Correlation of Active Anterior Rhinomanometry and Acoustic Rhinometry

When the data from AAR were correlated with those from AR, only a week correlation was found between total inspiratory resistance and NV 0–3 (r_s_ = −0.17, *p* = 0.008) (Table 7).

## 4. Discussion

Rhinomanometry was first introduced by Courtade in 1903 [18]. In 2016, the International Standardization Committee on the Objective Assessment of the Nasal Airway suggested that the logarithmic effective resistance measured by rhinomanometry was a parameter of high diagnostic relevance [19]. There are three methods in use with regards to rhinomanometry, AAR, passive anterior rhinomanometry and active posterior rhinomanometry. AAR is the method most frequently used, and it is considered to be simple, fast and well-tolerated.

According to the International Standardization Committee on the Objective Assessment of the Nasal Airway, a clinical classification for determining nasal obstruction in increments of 20% was provided for Caucasian noses based on 36,500 AAR measurements and 10,030 measurements of calculated total resistance. The cut-off value for total inspiratory resistance for 0–19% nasal obstruction was set at 0.42 [19]. In our study, based on subjects without any complaints of nasal obstruction, the 90th percentile value of total inspiratory resistance was 0.32 for Taiwanese males and 0.37 for Taiwanese females. We found that age did not have major impact on total inspiratory resistance but that gender may affect total inspiratory resistance when measured by AAR.

Acoustic rhinometry was first introduced by Hilberg in 1989 and was used to determine morphological changes in nasal airways [12]. The minimal cross-sectional area is the most commonly used parameter. MCA_1_ is located at the level of the nasal valve, and MCA_2_ is at the head of the inferior turbinate [18]. Nasal volume is another important parameter, and it is defined as the space between the opening plane of the device and a parallel plane at a defined distance from the opening plane [19].

Although there are no established standards and large inter-individual and ethnic variations exist, it has been mentioned that when the minimal cross-sectional area is less than 0.5 cm^2^, the sensation of nasal obstruction is reported as being severe [20]. In our study, the 10th percentile male average MCA_1_ was 0.52 in the age group of 20–39 years, 0.60 in the age group of 40–59 years and 0.62 in the age group of ≥60 years. The 10th percentile female average MCA_1_ was 0.41 in the age group of 20–39 years, 0.55 in the age group of 40–59 years and 0.54 in the age group of ≥60 years. The 10th percentile average MCA_2_ was 0.20 for Taiwanese males and 0.21 for Taiwanese females. Age and gender had an effect on average MCA_1_, but did not influence average MCA_2_. 

The effects of age and gender on nasal volumes measured by AR have rarely been investigated [14]. Our study shows that gender had an effect on average NV 0–3 but did not influence average NV 2–5. Males had a larger average NV 0–3 than females, while age did not influence nasal volumes. Similar findings were also reported by Samoliński et al. [21]. They found that after the age of 16, nasal cavities were bigger in males than in females. In contrast, Lindemann et al. reported MCA_1_, NV 0–2, MCA_2_ and NV 2–5 were statistically significantly higher in the older subjects compared with young adults, but they did not take gender into consideration [14].

The influence of age and gender on nasal patency has been investigated in several studies. Ganjaei et al. measured nasal volumes on computed tomography scans and found that older subjects had larger nasal volumes than young adults, but the clinical impact was unknown [22].

In addition to age and gender, many other factors might influence nasal patency. Body height has been shown to influence nasal patency, especially for children [23,24], but the effect of body height on nasal patency was not evaluated in this study. The nasal cycle has been known as spontaneous, cyclic congestion and decongestion in the two nasal cavities [25].AAR and AR have revealed a spontaneous fluctuation in nasal minimum cross-sectional area, volume and nasal resistances [26], but normal subjects are not usually aware of this phenomenon because the total nasal resistance usually remains fairly constant [27].

AAR and AR are the two most common methods for evaluating nasal airway patency. Although the correlation between objective assessment and subjective sensation of nasal patency remains uncertain [28,29], our study results show that the data of AAR did not correlate well with those of AR. 

## 5. Conclusions

AAR and AR are currently the most common tests for determining nasal function and, together, provide a clearer picture on nasal anatomy and physiology. In our study, we found that age did not influence total inspiratory resistance, while gender did have an effect on total inspiratory resistance. Alternatively, both age and gender had an effect on average MCA_1_ but did not influence average MCA_2_. Males had a larger average NV 0–3 than females, but age did not influence nasal volumes. Therefore, gender should be taken into consideration when measuring nasal patency using AAR and AR.

## Figures and Tables

**Figure 1 diagnostics-13-01235-f001:**
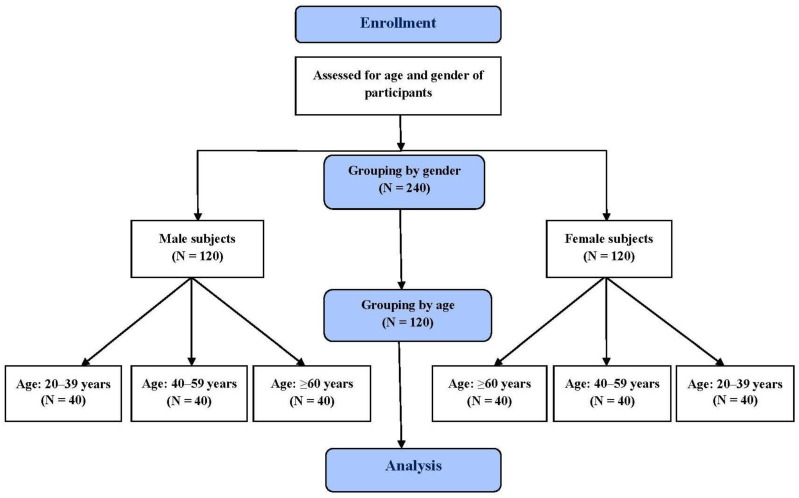
Participant flow chart.

**Figure 2 diagnostics-13-01235-f002:**
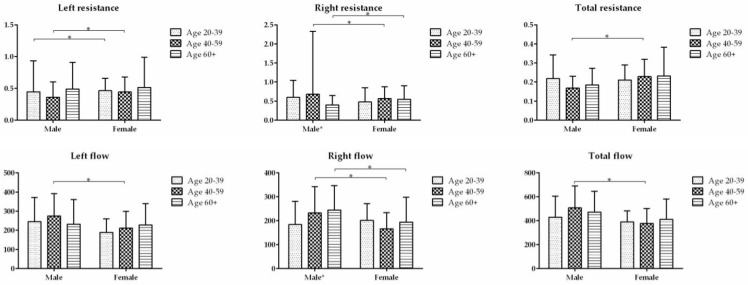
The nasal inspiratory resistance (Pa/cm^3^/s) and nasal flow (cm^3^/s) at a fixed pressure of 75 Pascal in 3 age groups were compared between the males and females. * indicates the difference is statistically significant (*p* < 0.05).

**Figure 3 diagnostics-13-01235-f003:**
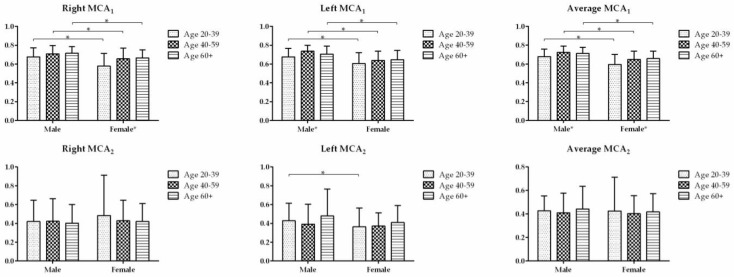
The first minimal cross-sectional area (MCA_1_) and second minimal cross-sectional area (MCA_2_) in three age groups were compared between the males and females. * indicates the difference is statistically significant (*p* < 0.05).

**Figure 4 diagnostics-13-01235-f004:**
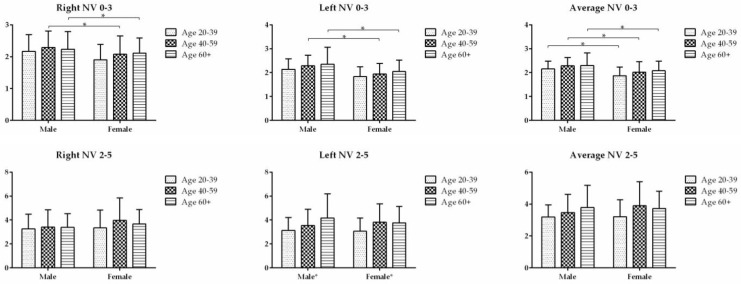
The NV 0–3 and NV 2–5 in three age groups were compared between the males and females. * indicates the difference is statistically significant (*p* < 0.05).

**Table 1 diagnostics-13-01235-t001:** Nasal inspiratory resistance (Pa/cm^3^/s) at a fixed pressure of 75 Pascal.

	Male		Female		*p*
	Mean	±SD	Mean	±SD	
Left resistance					
Age 20–39	0.45	±0.49	0.46	±0.20	0.025
Age 40–59	0.36	±0.24	0.44	±0.24	0.011
Age 60+	0.49	±0.42	0.51	±0.48	0.859
*p*	0.295		0.386		
Right resistance					
Age 20–39	0.60	±0.45	0.48	±0.38	0.148
Age 40–59	0.68	±1.65	0.57	±0.31	0.003
Age 60+	0.40	±0.25	0.55	±0.35	0.030
*p*	0.021		0.055		
Total resistance					
Age 20–39	0.22	±0.12	0.21	±0.08	0.266
Age 40–59	0.17	±0.06	0.23	±0.09	0.001
Age 60+	0.18	±0.09	0.23	±0.15	0.088
*p*	0.267		0.540		

**Table 2 diagnostics-13-01235-t002:** Nasal flow (cm3/s) at a fixed pressure of 75 Pascal.

	Male		Female		*p*
	Mean	±SD	Mean	±SD	
Left flow					
Age 20–39	245.67	±126.30	188.94	±70.58	0.054
Age 40–59	274.68	±117.17	210.98	±88.05	0.011
Age 60+	231.70	±128.63	227.55	±111.96	0.881
*p*	0.265		0.231		
Right flow					
Age 20–39	183.79	±97.18	201.54	±69.51	0.144
Age 40–59	233.06	±109.24	165.91	±68.12	0.003
Age 60+	244.68	±102.18	194.33	±103.80	0.027
*p*	0.013		0.061		
Total flow					
Age 20–39	429.44	±174.80	389.99	±93.37	0.256
Age 40–59	507.74	±183.40	376.88	±126.19	0.001
Age 60+	471.37	±175.22	411.27	±170.99	0.145
*p*	0.247		0.630		

**Table 3 diagnostics-13-01235-t003:** First minimal cross-sectional area (MCA_1_,cm^2^).

	Male		Female		*p*
	Mean	±SD	Mean	±SD	
Right MCA_1_					
Age 20–39	0.68	±0.10	0.58	±0.13	0.001
Age 40–59	0.71	±0.09	0.66	±0.12	0.018
Age 60+	0.72	±0.07	0.66	±0.09	0.008
*p*	0.072		0.003		
Left MCA_1_					
Age 20–39	0.68	±0.09	0.61	±0.12	0.005
Age 40–59	0.74	±0.06	0.64	±0.10	<0.001
Age 60+	0.71	±0.08	0.65	±0.10	0.005
*p*	0.013		0.234		
Average MCA_1_					
Age 20–39	0.68	±0.08	0.59	±0.11	<0.001
Age 40–59	0.72	±0.07	0.65	±0.09	<0.001
Age 60+	0.71	±0.07	0.66	±0.08	0.002
*p*	0.012		0.019		

**Table 4 diagnostics-13-01235-t004:** Second minimal cross-sectional area (MCA_2_, cm^2^).

	Male		Female		*p*
	Mean	±SD	Mean	±SD	
Right MCA_2_					
Age 20–39	0.42	±0.23	0.48	±0.43	0.958
Age 40–59	0.42	±0.24	0.43	±0.22	0.977
Age 60+	0.40	±0.20	0.42	±0.19	0.679
*p*	0.979		0.781		
Left MCA_2_					
Age 20–39	0.43	±0.19	0.36	±0.20	0.046
Age 40–59	0.39	±0.21	0.37	±0.14	0.923
Age 60+	0.48	±0.28	0.41	±0.18	0.358
*p*	0.359		0.251		
Average MCA_2_					
Age 20–39	0.42	±0.13	0.42	±0.29	0.097
Age 40–59	0.41	±0.17	0.40	±0.15	0.707
Age 60+	0.44	±0.19	0.42	±0.16	0.528
*p*	0.570		0.417		

**Table 5 diagnostics-13-01235-t005:** Nasal volume (cm^3^) between the tip of the nosepiece and 3.0 cm into the nasal cavity (NV 0–3).

	Male		Female		*p*
	Mean	±SD	Mean	±SD	
Right NV 0–3					
Age 20–39	2.17	±0.53	1.90	±0.48	0.023
Age 40–59	2.29	±0.51	2.08	±0.57	0.035
Age 60+	2.23	±0.55	2.11	±0.48	0.114
*p*	0.501		0.127		
Left NV 0–3					
Age 20–39	2.14	±0.44	1.83	±0.41	0.001
Age 40–59	2.28	±0.45	1.94	±0.44	0.001
Age 60+	2.35	±0.71	2.05	±0.48	0.054
*p*	0.213		0.109		
Average NV 0–3					
Age 20–39	2.15	±0.33	1.87	±0.36	<0.001
Age 40–59	2.29	±0.35	2.01	±0.44	0.001
Age 60+	2.29	±0.53	2.08	±0.40	0.033
*p*	0.253		0.068		

**Table 6 diagnostics-13-01235-t006:** Nasal volume (cm^3^) of the nasal cavity between 2.0 and 5.0 cm from the tip of the nosepiece (NV 2–5).

	Male		Female		*p*
	Mean	±SD	Mean	±SD	
Right NV 2–5					
Age 20–39	3.26	±1.22	3.34	±1.50	0.935
Age 40–59	3.40	±1.44	3.97	±1.88	0.218
Age 60+	3.38	±1.14	3.67	±1.21	0.459
*p*	0.787		0.168		
Left NV 2–5					
Age 20–39	3.12	±1.09	3.06	±1.11	0.655
Age 40–59	3.53	±1.36	3.81	±1.55	0.532
Age 60+	4.17	±2.01	3.76	±1.38	0.557
*p*	0.019		0.012		
Average NV 2–5					
Age 20–39	3.19	±0.76	3.20	±1.06	0.784
Age 40–59	3.47	±1.15	3.89	±1.50	0.256
Age 60+	3.78	±1.40	3.72	±1.08	0.754
*p*	0.222		0.073		

**Table 7 diagnostics-13-01235-t007:** Correlation of active anterior rhinomanometry and acoustic rhinometry.

	Nasal Inspiratory Resistance					
	Left		Right		Total	
	r_s_	*p* Value	r_s_	*p* Value	r_s_	*p* Value
MCA_1_	−0.07	0.264	0.00	0.994	−0.06	0.345
MCA_2_	−0.16	0.014	−0.23	<0.001	−0.12	0.073
NV 0–3	−0.15	0.020	−0.24	<0.001	−0.17	0.008
NV 2–5	−0.13	0.048	−0.11	0.100	−0.11	0.100

## Data Availability

Not applicable.

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
