# Peer review of "Influence of Age and Gender on Nasal Airway Patency as Measured by Active Anterior Rhinomanometry and Acoustic Rhinometry"

_diagnostics, 2023, doi:10.3390/diagnostics13071235_

Round 1

Reviewer 1 Report

see uploaded file

This study in healthy Taiwanese inviduals investigated the effects of age and gender on nasal geometry and resistance using rhinomanometry and acoustic rhinometry. The main finding is that gender but not age was correlated with outcomes with women having smaller cross-sectional areas and nasal volumes than men.

The study is of potential clinical relevance as interpretation of rhinomanometry and acoustic rhinometry are commonly used to objectively assess nasal patency in patients with nasal symptoms as an adjunct to guide treatment. Nevertheless, the report might be improved by expanding the statistical analysis, improving the data presentation and modifying the discussion as suggested below.

Specific comments

Methods

Recruitment: please, specify how participants were recruited.

Please include a statement about ethics committee approval and participant consent.

Data analysis: I recommend to use regression models (either linear or logistic) to more comprehensively and powerfully evaluate the effects of age and gender on outcomes while controlling for potential confounders. This approach could also be used to create prediction equations. I strongly suggest to include body height into the analysis (compute height-adjusted values) as I suspect that differences in height between men and women might explain some of the differences in nasal geometry.

Results

Please, provide a participant flow chart.

I suggest to graphically illustrate the main results. This will facilitate understanding of the main trends at a glance.

Redundancy among text and tables should be kept at a minimum.

Figures 1 and 2 could be omitted or replaced by schematic drawings that illustrate the measurement principles. Alternatively, just providing a literature reference to these widely used measurement methods would be appropriate as well.

Table 2, title: mention that the flow was at 75 hPa

Tables: provide measurement units

Discussion

The first paragraphs of the discussion (lines 221-239) would be better placed in the introduction and could be reduced in length.

The nasal cycle or its disturbances might influence symptoms and objective findings. Please, discuss whether the nasal cycle might bias results of a single measurement of rhinomanometry and acoustic rhinometry.

The statement in the last paragraph of the discussion, L271-273: should be reconsidered ‘’…when evaluating nasal airway patency, it is better to analyze both AAR and AR data simultaneously in order to reach a more accurate conclusion, in addition to subjective assessment…’’ This statement is not supported by the results of the current study. Please, omit or corroborate by reference to relevant data.

Conclusions: Please provide a statement on the potential clinical relevance of the findings.

Reviewer 2 Report

Authors wished to investigate the influence of age and gender on nasal airway patency using both active anterior rhinomanometry and acoustic rhinometry. They found that nasal airway patency was not affected by age, while the females were shown to have wider nasal passage than males.

 Critical point:

Authors should discuss why their results are in contrast with the literature, as Lindemann et al. reported that both the cross-sectional areas of the nasal cavities and nasal volumes were significantly higher in the elderly than in young subjects. 

More recently Ganjaei KG er al (Rhinology 2019), by using CT scan, reported that older subjects  have a global increase in intranasal volumes and diffuse bone density loss.

I think the reason of the difference may be in the older age range examined by Lindeman (61-84) and by Ganjaei (80-99 yrs).
